# Return to Driving after Elective Foot and Ankle Surgery: A Systematic Review

**Alexander Lundy** [1,2,*] , **Andres Piscoya** [1,2] , **Daniel Rodkey** [1,2] , **Michael Bedrin** [1,2] and **Tobin Eckel** [1,2]

1   Walter Reed National Military Medical Center, 4494 Palmer Rd N, Bethesda, MD 20814, USA; apiscoya23@gmail.com (A.P.); daniel.rodkey@gmail.com (D.R.); michaelbedrin@gmail.com (M.B.); tobin.t.eckel.mil@mail.mil (T.E.)

2   Orthopaedic Surgery, Department Uniformed Services, University of the Health Sciences, 4301 Jones Bridge Rd, Bethesda, MD 20814, USA

\*   Correspondence: alexelundy@gmail.com

**Abstract:** (1) Background: This systematic review summarizes the available studies investigating when it is safe for most patients to return to driving and when to modify for individual patients following elective foot and ankle procedures. (2) Methods: A systematic review of the literature was performed using three different electronic databases to identify English-language studies from 1999 to present that investigate the return to driving after right-sided elective foot and ankle procedures. (3) Results: A total of eight studies met inclusion criteria. All the studies investigated brake reaction time (BRT) as measured by a driving simulator as their primary outcome. Patients undergoing right ankle or subtalar arthroscopy should wait 2 weeks to drive, after total ankle arthroplasty or corrective hallux valgus surgery patients should wait 6 weeks, and the appropriate time to return to driving after ankle arthrodesis is still uncertain. Additionally, various clinical factors can be used to predict who may still be unfit to drive past the usual length of time. (4) Conclusions: The recommendations from these reviewed studies can guide physicians when counseling their patients on when they can expect to safely return to driving after a specific elective foot and ankle procedure. However, these recommendations should be tailored to the patient specifically based upon how they are doing clinically.

**Keywords:** foot and ankle surgery; return to driving; ankle arthroscopy; hallux valgus; ankle arthroplasty; ankle arthrodesis

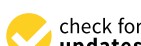



## 1. Introduction

Patients often inquire about when they are safe to return to driving after lower-extremity surgery. Interestingly, very little guidance from government and professional organizations exists on how to answer these questions. This places surgeons in a difficult position because being able to accurately counsel a patient on when they can expect to safely return to driving after a surgery is important in building an effective patient–physician relationship. This was highlighted by a recent study surveying orthopaedic surgeons in which 76% of surgeons did not have a consistent return-to-driving policy [1]. The overall lack of clarity potentially leaves surgeons and their patients in a compromised medico-legal position [2]. Egol et al. were among the first who attempted to answer this question for patients with ankle fractures and other complex lower-extremity trauma [3,4]. Since then, several studies have begun to explore how specific elective foot and ankle cases such as hallux valgus surgery, total ankle arthroplasty, ankle arthroscopy, and mid- and hind foot arthrodesis affect return to driving. Previous review papers have focused on return to driving after a wide variety of orthopedic procedures [5,6] but, to our knowledge, there has not been a systematic review of literature evaluating outcomes and return to driving after elective foot and ankle surgery.

Assessment of safe return to driving after foot and ankle surgery is difficult and is often dependent on the individual and procedure. Although there are many different variables that go into safe driving, one that would obviously be affected by foot and ankle surgery would be the ability to perform an emergency stop. The transportation literature defines emergency braking as the time from recognition by the driver to the time it takes for the vehicle to stop and within this process is the brake reaction time (BRT), which is the time between driver recognition to when the brake pedal is depressed [7]. The average, healthy person should have a brake reaction time between 700 and 750 milliseconds [7]. As such, surgeons have attempted to answer when patients can safely return to driving by using driving simulators to measure parameters such as brake reaction time. To account for subjective variables that may affect driving ability, studies have begun to utilize subjective patient reported outcomes to determine if correlations exist between measured braking data and how patients are recovering clinically to better assess real-world driving safety. To provide surgeons guidance, we systematically queried the literature, while performing quality appraisal of the data, to answer two main questions: 1. When patients can safely return to driving after elective foot and ankle surgery? 2. Are there ways to predict when a specific patient is unsafe to drive post-operatively?

## 2. Materials and Methods

### 2.1. Literature Review

This study was performed in accordance with the PRISMA guidelines without a previously registered protocol or any sources of external funding. A literature review of the PubMed, Embase, and CINAHL electronic databases was performed with the following key words and Boolean operators: "braking time OR driving" AND "foot surgery OR ankle surgery" with filters for English Language, between the years of 1999 and 2020, and full-text articles. Two independent reviewers (AL and AP) assessed the titles and abstracts separately for relevancy. Abstracts that were considered for inclusion in this investigation were kept for full manuscript review. Inclusion and exclusion criteria were then applied to these articles for analysis.

Qualitative assessment of the included studies was performed using the Methodological Index for Nonrandomized Studies (MINORS) checklist by the two reviewing authors. The MINORS scores from each author were then averaged together to produce a single score for each study. For the MINORS criteria, a non-comparative study can receive an ideal score of 16 and a comparative study can receive an ideal score of 24. The level of evidence of each study was decided upon jointly by the two reviewing authors. If a level of evidence was stated in the article by the authors, then this level was accepted unless there was an obvious reason to change it.

### 2.2. Eligibility

The following inclusion criteria were applied to the selected studies: (1) primary studies on elective foot and/or ankle procedures; (2) studies measuring braking time post-operatively. Exclusion criteria were: (1) abstract-only studies, (2) non-elective foot and/or ankle surgery, (3) amputation related studies.

### 2.3. Data Extraction and Analysis

The two authors independently extracted data from the studies that met criteria and grouped the data into 5 distinct categories: 1. Study details 2. Population 3. Control group 4. Method of evaluation 5. Results. Study details focused on the level of evidence of the study, type of study performed, and the MINORS score. Population was the characteristics of the experimental group. Control group was the characteristics of the control group if it was a comparative study. Method of evaluation included the various outcome measures for each study, i.e., brake reaction time, VAS scores, driving readiness surveys, or driving habit surveys. The results were the empirical data collected for each of the outcome measures.

## 3. Results

### 3.1. Study Details

Our search of the electronic databases produced 790 unique results. After review, eight studies met inclusion for this systematic review (Figure 1). Three studies were level II, two studies were level III, and three studies included in this review were level IV.

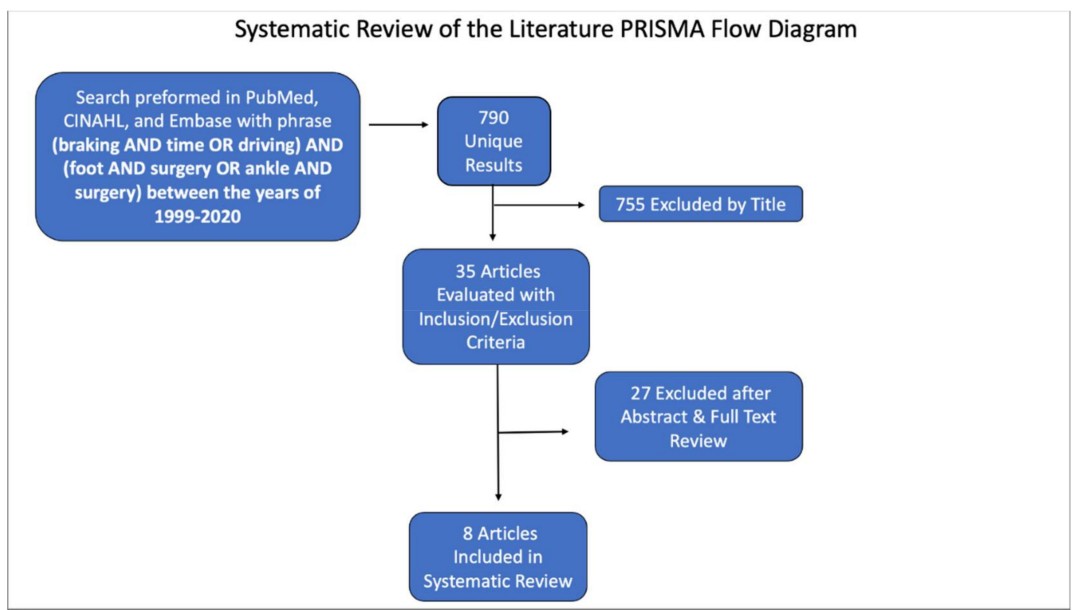

**Figure 1.** PRISMA Flow Diagram.

### 3.2. Population and Control Group

There were four different elective procedures analyzed between the eight included studies: two studies dealt with ankle arthroscopy, one with total ankle arthroplasty, two with ankle arthrodesis, and three with hallux valgus. All but two studies used a control group of volunteers, and the highest MINORS score went to the Holt et al. study (Table 1).

**Table 1.** MINORS Results.

| Study | Procedure | Experimental Group | Control Group | Average MINORS Score |
|---|---|---|---|---|
| Dammerer et al. [8] | Hallux valgus | 42 patients | No control group | 13 |
| Holt et al. [9] | Hallux valgus | 28 patients | 28 age-, gender-, and driving-frequency-matched volunteers | 19.5 |
| Jeng et al. [10] | Ankle arthrodesis | 10 patients | 10 age-matched volunteers | 16 |
| Liebensteiner et al. [11] | Ankle arthroscopy | 19 patients | No control group | 15 |
| McDonald et al. [12] | Hallux valgus | 60 patients | 20 age- and gender-matched volunteers | 18 |
| McDonald et al. [13] | Total ankle arthroplasty | 59 patients | 20 age- and gender-matched volunteers | 19 |
| Schwienbacher et al. [14] | Ankle arthrodesis and other foot joint arthrodesis | 12 patients with ankle arthrodesis 12 patients with other foot joint arthrodeses | 17 volunteers | 16 |
| Sittapairoj et al. [2] | Ankle arthroscopy | 17 patients | 19 age-matched volunteers | 17 |

### 3.3. Method of Evaluation

All eight studies used brake reaction time (BRT) as one of their primary outcomes. Most of the studies defined the BRT as the time between the presentation of a stimulus to the time when the driver engages the brake pedal. Depending upon the specific set-up, some studies required a certain force to be on the brake pedal or a certain depth of depression before time stopped [2,12,13], while others simply counted engagement of the

pedal [9–11]. Schweinbacher et al. defined total brake reaction time as the summation of reaction time and movement time, thereby allowing them to differentiate if one phase was more significantly affected [14].

Subjective patient-reported outcomes were also considered in many of the studies. Surveys assessing driver readiness [12,13] and driving frequency [11] were employed evaluate driving habits. McDonald et al. also included VAS and AOFAS-Hallux valgus scores [13]. Liebensteiner et al. looked at ankle osteoarthritis scale and AOFAS ankle/hindfoot scales as secondary outcomes [11].

### 3.4. Objective Observer-Reported Outcomes

The primary objective outcome for all studies was BRT. At 2 weeks post operatively patients undergoing right ankle or subtalar arthroscopic procedures have no significant difference in BRT when compared to their pre-operative results or to a healthy control group (Table 2) [2,11]. Interestingly, patients who drove more often had significantly faster BRT than patients who drove infrequently at the 2-day postoperative mark; however, this difference equalized at the 2-week mark. For patients undergoing right ankle arthroplasty, 92% of them had a BRT that was not significantly different to the control group at 6 weeks post-operatively. The 8% that had significantly longer BRT also had a significantly higher VAS score and reduced plantarflexion when compared to the passing group (Table 3) [13]. Patients who had an ankle arthrodesis had significantly delayed BRT compared to a healthy control group even past 1-year post-operatively [8,10]. When breaking down the phases of BRT, movement time, which was defined as the interval between the start of force decrease off the accelerator and the start of force application on the brake pedal, was significantly slower in patients who underwent ankle arthrodesis while reaction time was not significantly different [14]. Patients who have an arthrodesis of any joint in the foot have no significant difference in BRT at 6 months post-operatively compared to controls (Table 4) [14]. Patients after corrective hallux valgus surgery generally returned to their own pre-operative BRT baseline at 6 weeks post-operatively, but they can still show a significant difference compared to a healthy control at that time (Table 5) [8,9,12]. In one study, 85% of the post-operative patients achieved the passing threshold, which was set by the slowest BRT in the control group, by 6 weeks. The driver readiness survey used in this study was strongly predictive of who would pass at the 6-week mark and who would fail [12]. The study by Holt et al. showed no significant difference in pre-operative BRT compared to 6 weeks post-operative BRT. However, both time points were significantly slower in operative patients when compared to healthy controls [9]. Similarly, one of the other studies showed that at 6 weeks, although the operative group showed no significant difference in their BRT compared to their pre-operative marks, they were significantly slower when compared to the healthy control group [9]. Of note, one study investigated the effects of post-operative foot orthoses and found that it significantly reduced BRT at all time points [8].

**Table 2.** Brake Reaction Time Results: Ankle Arthroscopy.

| Study | Control Mean BRT (ms) | 2-Week Mean BRT (ms) | *p*-Value | 6-Week Mean BRT (ms) | *p*-Value |
|---|---|---|---|---|---|
| Liebensteiner et al. [11] | 606, pre-op baseline | 606 | Not significant ♦ | 596 | Not significant ♦ |
| Sittapairoj et al. [2] | 550, healthy control | 570 | $p = 0.84$ | N/A | N/A |

♦ Exact value not given in study.

After an ankle arthrodesis procedure, the functionality of a patient can change secondary to the lack of range of motion. With reference to braking, an ankle arthrodesis forces the patient to concentrate the force used to depress the brake pedal into their forefoot, while healthy controls distribute the force between their forefoot and midfoot [10].

Holt et al. included a step test in their objective outcomes where the subject stepped over a 5 cm block with their right foot, while seated, as many times as possible within

10 s [9]. When compared to their pre-operative scores, the patients significantly slowed at 2 weeks post-operatively, but then significantly improved over their pre-operative score at 6 weeks post-operatively. This was consistent with their finding of BRT significantly improving from the pre-operative baseline at the 6-week mark [9].

**Table 3.** Brake Reaction Time Results: Ankle Arthroplasty.

| Study | Control Mean BRT (ms) | 6-Week Mean BRT (ms) | *p*-Value |
|---|---|---|---|
| McDonald et al. [13] | 547, healthy control | 620 passing group * (*n* = 54) <br> 1120 failing group * (*n* = 5) | <0.001 @ |

* Passing group refers to the members of the experimental group who achieved a BRT that was faster than 850'milliseconds, while the failing group refers to the members of the experimental group who had a BRT slower than 850 milliseconds. @ *p*-value when comparing passing group to the failing group at 6 weeks post-operatively.

**Table 4.** Brake Reaction Time Results: Ankle Arthrodesis.

| Study | Control Mean BRT (ms) | 6-Month Mean BRT (ms) | *p*-Value | 12-Month Mean BRT (ms) | *p*-Value |
|---|---|---|---|---|---|
| Schweinbacher et al. [14] | 475.7, healthy control | 596.2 other foot joint arthrodesises <br> 729.7 ankle arthrodesis | 1 μ <br> 0.026 μ | N/A | N/A |
| Jeng et al. [10] | 330, healthy control | N/A | N/A | 420 | 0.03 λ |

μ *p*-value when comparing 6-month post-operative BRT to control BRT. λ *p*-value when comparing 12-month post-operative BRT to control BRT.

**Table 5.** Brake Reaction Time Results: Hallux Valgus Correction.

| Study | Control Mean BRT (ms) | 2-Week Mean BRT (ms) | *p*-Value | 6-Week Mean BRT (ms) | *p*-Value |
|---|---|---|---|---|---|
| Dammerer et al. [8] | 712, pre-op baseline with no orthosis | 804 with Hallux Valgus Shoe <br><br> 841 with Forefoot Relief Shoe | <0.001 # <br><br> <0.001 # | 750 with Hallux Valgus Shoe <br> 811 with Forefoot Relief Shoe <br> 769 with no orthosis | 0.003 δ <br> <0.001 δ <br> <0.001 δ |
| Holt et al. [9] | 806, pre-op baseline | 850 | 0.791 # | 684 | <0.001 δ |
| McDonald et al. [12] | 550, healthy control | N/A | N/A | 640 passing group * (*n* = 51) <br> 1360 failing group * (*n* = 9) | 0.02 @ |

# *p*-value when comparing 2-week post-operative BRT to control BRT. δ *p*-value when comparing 6-week post-operative BRT to control BRT. * Passing group refers to the members of the experimental group who achieved a BRT that was faster than 850 milliseconds, while the failing group refers to the members of the experimental group who had a BRT slower than 850 milliseconds. @ *p*-value when comparing passing group to the failing group at 6 weeks post-operatively.

### 3.5. Subjective Patient-Reported Outcomes

Specific to these driving studies, subjective patient-reported surveys were created to try to characterize the different driving habits of each patient. McDonald et al. used a 4-question driver readiness survey in their studies on patients after corrective hallux valgus surgery and after ankle arthroplasty [12,13]. This survey asked patients four questions related to how the patient perceives their braking ability. In the analysis of driving after hallux valgus surgery, the driver readiness survey was the most reliable way to predict who would achieve a failing BRT [12]. In the study on driving after ankle arthroplasty, a score of 10–15 points was considered a passing grade on the driver readiness survey. The five patients who failed to achieve a passing BRT also received failing grades on this survey at the 6-week follow-up. Furthermore, better BRT was correlated with better scores on the survey [13]. Holt et al. categorized their patients based upon how frequent they drove. Frequent drivers were found to have significantly reduced BRT compared to infrequent drivers at baseline pre-operatively (708 milliseconds ± 106 m vs. 958 milliseconds ± 317, respectively, $p = 0.006$). This significant difference persisted at the 6-week mark with frequent drivers having significantly faster BRT (646 milliseconds ± 124 vs. 794 ± 184, respectively, $p = 0.017$) [9].

McDonald et al. and Liebensteiner et al. used other patient reported outcomes to better asses why patients would do poorly on the driving simulator [11–13]. At 6 weeks post-operatively, patients who achieved a passing BRT after corrective hallux valgus surgery were significantly more likely to have a lower VAS score than some who failed. Meanwhile, age, AOFAS-Hallux Valgus score, and range of motion showed no significant predictive ability [12]. Similarly, patients after right ankle arthroscopy who were doing worse clinically had a strong correlation with slower BRT [11].

## 4. Discussion

The ability to tell a patient considering an elective foot and ankle procedure accurately and concisely when they will be able to safely drive again is incredibly important. State laws and regulations provide no real guidance to patients or surgeons regarding this matter [15]. Yet, instruction on return to driving should be a part of a surgeon's post-operative protocol just like time to weight bearing, wound care, and immobilization. As most surgeons do not seem to have a consistent driving policy, systematic reviews of the available literature can provide an excellent starting point [1].

Based upon this systematic review, we can make the following recommendations for when a typical patient can return to driving after certain elective foot and ankle procedures, with some important caveats. Patients after right ankle or subtalar arthroscopy can usually return to driving at 2 weeks post-operatively [2,11]. Most patients after right total ankle arthroplasty can return to driving at 6 weeks post-operatively; however, patients who are in above-average pain or who have below-average range of motion should be cautioned against driving until at least 8–9 weeks post-operatively [13]. After corrective right hallux valgus surgery, most patients can expect to be safely driving again at 6 weeks post-operatively; however, if patients are not doing as well clinically or feel personally unsure about their ability to drive safely then they should likely abstain for another 2 weeks [8,9,12]. Finally, although we did not find a specific post-operative threshold when patients after right ankle arthrodesis can return to driving, it was interesting that the two related studies showed that brake reaction time was significantly longer in this cohort than in healthy people, even more than one year post-operatively [10,14]. However, if less than 700–750 milliseconds for BRT is used as the threshold for driving safely [7], then at 1 year post-operatively, patients with right ankle arthrodesis were below the threshold [10]. At 6 months post-operatively, patients fell in the middle of that threshold, thus making driving risky [14].

Brake reaction time is an important objective measurement that can predict when the average patient is safe to return to driving, but subjective patient outcomes help to predict when a specific patient can return to driving. As a physician cannot put every post-operative patient through a driving simulator test, many of the reviewed studies attempt to introduce other methods of predicting which individual patients would theoretically fail the test. As one would expect, patients who are having more pain or have reduced range of motion have a higher likelihood of performing poorly. In addition, patients should be included in the discussion of whether they can safely return to driving because they know their personal limitations and driving habits best. Therefore, aside from the few definite rules that patients should not drive while on narcotics or in any form of right lower-extremity immobilization, the recommendation of when to return to driving is still variable based upon the specific patient.

There are several limitations with this study. First, although there is increasing literature on returning to driving in the orthopedics, this systematic review is still limited by the amount and quality of existing literature specific to elective foot and ankle procedures. The studies on return to driving after corrective hallux valgus surgery and ankle arthroscopy include a variety of specific procedures and a variety of indications all grouped under one category. For example, a patient undergoing ankle arthroscopy and debridement for anterior impingement is grouped in with a patient having microfracture [2,11], both of which have different post-operative rehab protocols. Similarly, there are several different types of

osteotomies and soft tissue procedures that may be included in corrective hallux valgus surgery. This could certainly affect the accuracy of return to driving recommendations for specific patients.

Then, the primary outcome of braking studies, brake reaction time, has several limitations itself. First, it has inherently limited real-world application. There are some considerations on how to change this, as one study that looked at return to driving after right ankle open reduction and internal fixation had patients go through an on-road exercise in the style of the national driver's license examination as the final test to determine if the patient would be able to safely return to driving post-operatively [16]. While this method would be more time consuming and less cost-effective, it would reduce the limitations of a driving simulator. Another limitation of BRT is that each study uses a different simulator to measure it and use slightly different criteria to define it. Although all of the studies generally consider BRT as the time it takes between introduction of a stimulus to the time it takes to reach the brake pedal, some of them mark the time as soon as the brake pedal is engaged, while others mark the time after a specific depth or force on the pedal has been achieved. When comparing studies of the same patient population, this may limit the generalizability. Next, there are a variety of control groups used by each study in this review. If you use the BRT of a healthy volunteer group, there is a wide range reported in the literature. This complicates recommendations, as many studies base when they recommend returning to driving off when the post-operative group's BRT equals the healthy control. Standardizing what is considered a "normal" BRT would greatly help with the consistency of return to driving recommendations. For example, the two studies on patients with ankle arthrodesis note that the operative group never regains the BRT that the healthy control group has, which taken out of context would suggest that patients after ankle arthrodesis are unsafe to drive [10,14]. If 700–750 milliseconds were used as a standard instead, the recommendation would become completely different. The variability even in the BRT of the healthy control population also raises the question as to what may predispose someone to having a faster brake reaction time. Some of the studies in this review attempt to answer this by comparing groups who drive frequently to those who drive less frequently [9]. Other studies use a patient's pre-operative baseline as the control to remove this variable completely. Regardless of the method, having a standardized control population or an accepted safe threshold for BRT could potentially allow for more accurate generalizations, and therefore recommendations, across studies. Then, the subjective patient outcomes could act as a guide for specific situations.

### 5. Conclusions

This systematic review is, to our knowledge, the first report in the literature to compile and analyze data of clinical studies measuring braking characteristics after elective foot and ankle surgery. Our goal is to provide physicians guidance when counseling patients undergoing elective foot and ankle surgeries on when they can reliably expect to return to driving. Even if the ideal time for each individual is difficult to determine, these general recommendations can provide reasonable expectations prior to the procedure. Ultimately, it is up to the patient to decide when they feel safe to return to driving. Future studies should include other elective foot and ankle procedures, such as lateral ligament reconstruction, to promote further understanding of how driving post-operatively is affected.

**Author Contributions:** Conceptualization, T.E. and A.P.; methodology, A.P. and A.L.; investigation A.P. and A.L.; writing—original draft preparation A.L.; writing—review and editing D.R., T.E. and M.B.; supervision T.E. All authors have read and agreed to the published version of the manuscript.

**Funding:** This research received no external funding.

**Institutional Review Board Statement:** This study did not require IRB approval because it is a systematic review.

**Informed Consent Statement:** Not applicable.

**Data Availability Statement:** Data sharing not applicable.

**Conflicts of Interest:** The authors declare no conflict of interest.

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
