# Peer review of "Return to Driving after Elective Foot and Ankle Surgery: A Systematic Review"

_2673-4036, doi:10.3390/osteology2030014_

Round 1

Reviewer 1 Report

Comments and Suggestions for Authors
  1. There are too few papers analyzed, making it difficult for this systemic review to effectively analyze the problems it wants to answer, and the content is mostly directly quoting the findings of reference papers.

  2. The recommendations for “come back to drive” after joint surgery are directly described without effective analysis (such as meta analysis), and it is not clear what criteria the author accepts or rejects the results from collected papers.

  3.  English grammar needs to be improved, especially in Results and Abstracts.

  4.  The table 1-3 are very misaligned and character styles are not correct.

Author Response

 Thank you very much for taking the time to consider our study and provide feedback.

  1. As we mentioned in the discussion, we agree that there are too few papers on this subject. Unfortunately, after our literature review these were the only studies on this subject matter that met our inclusion criteria. We hope that reviews like ours will encourage more driving studies in the future.
  2. Similarly, we felt that because of the limited number of studies and some of the heterogeneity between the studies we could not effectively or reliably turn this into a meta analysis. Therefore we decided to keep it as a systematic review. 
  3. I went through the results and discussion sections again with a co-author to fix the grammatical errors. We believe that they have been addressed. 
  4. I have corrected the alignment and the font type of all of the tables. Furthermore I have separated out the results table to make it easier to read and understand. 

Reviewer 2 Report

Comments and Suggestions for Authors

This is a nice work. There are some limitations but these are adequately taken care of in the text. I have som comments.

Table 1 is not necessary, better to state this in text. I do´nt understand the meaning of the numbers in brackets after the author names in Table 2. Table 3 is impossible. It has to be compressed and diminished in some way. Line 215 - 232 is a result paragraph in the Discussion section already stated in the Outcome section. In Diskussion the results should be discussed not stated once again.

Author Response

  1. I have taken out table 1 and edited some of the text to include the same information.
  2. The bracketed numbers after the author's names in the tables are the numbers associated with our bibliography. We wanted to make sure each article was properly cited. 
  3. We have broken up Table 3 into separate tables, which are hopefully much easier to read and understand. 
  4. We wanted that part of the discussion to be a concise list of all our recommendations based upon the reviewed literature. We didn't want to take it out because we feel that it nicely summarizes the recommendations based upon the literature as a whole. 

Thank you very much for taking the time and consideration to review our work. 

Round 2

Reviewer 1 Report

Comments and Suggestions for Authors

  1. The authors of this manuscript have fixed some problem including the table spelling and the editing. However, there are still present some editing unfamiliar to medicine audiences. For example, Line 14-16: "patients undergoing right ankle or subtalar arthroscopy should wait 2 weeks to drive, after total ankle arthroplasty patients should wait 6 weeks, after corrective hallux valgus surgery patients should wait 6 weeks" may change the type of the surgery as the subject like "right ankle or subtalar arthroscopy need the patients wait 2 weeks to drive after surgery, while, both total ankle arthroplasty and corrective hallux valgus surgery need the patients wait up to 6 weeks after surgery.

2. Line 146: "the operative patient" is strange to medicine audiences.

3. Line 219-221 "Based upon this systematic review, we can make the following recommendations for when the average patient can return to driving after specific elective foot and ankle procedures, with some important caveats." Both "the average patient"  and  "for when .. can return"  are not correct editing.

4. This manuscript can be published after fixing unfamiliar English editing for medicine audiences.

Author Response

  1. I have condensed this line to the following: "Patients undergoing right ankle or subtalar arthroscopy should wait 2 weeks to drive, after total ankle arthroplasty or corrective hallux valgus surgery patients should wait 6 weeks, and the appropriate time to return to driving after ankle arthrodesis is still uncertain."
  2. I understand how that wording is strange. I have changed the sentence structure around in order to make it easier to understand. 
  3. I have edited this sentence to the following: "Based upon this systematic review, we can make the following recommendations for when a typical patient can return to driving after certain elective foot and ankle procedures, with some important caveats."

Thank you again for your consideration and suggestions. 
